# Incidence of Type 1 Diabetes Mellitus and Characteristics of Diabetic Ketoacidosis in Children and Adolescents during the First Two Years of the COVID-19 Pandemic in Vojvodina

**DOI:** 10.3390/medicina58081013

**Published:** 2022-07-28

**Authors:** Ivana Vorgučin, Marijana Savin, Đurđina Stanković, Dejan Miljković, Tatjana Ilić, Danijela Simić, Mirjana Vrebalov, Borko Milanović, Nenad Barišić, Vesna Stojanović, Gordana Vijatov-Đurić, Katarina Koprivšek, Gordana Vilotijević-Dautović, Jelena Antić

**Affiliations:** 1Institute for Child and Youth Health Care of Vojvodina, 21000 Novi Sad, Serbia; marijana.savin.md@gmail.com (M.S.); borko.milanovic@mf.uns.ac.rs (B.M.); nenad.barisic@mf.uns.ac.rs (N.B.); vesna.stojanovic@mf.uns.ac.rs (V.S.); gordana.vijatov-djuric@mf.uns.ac.rs (G.V.-Đ.); katarina.koprivsek@mf.uns.ac.rs (K.K.); gordana.vilotijevic-dautovic@mf.uns.ac.rs (G.V.-D.); jelena.antic@mf.uns.ac.rs (J.A.); 2Faculty of Medicine, University of Novi Sad, 21000 Novi Sad, Serbia; dejan.miljkovic@mf.uns.ac.rs; 3General Hospital Subotica, 24000 Subotica, Serbia; drtatjanailic@gmail.com; 4General Hospital Sombor, 25000 Sombor, Serbia; danijelaso@live.com; 5General Hospital Zrenjanin, 23000 Zrenjanin, Serbia; vrebalovm@gmail.com

**Keywords:** COVID-19, pandemic, type 1 diabetes mellitus, diabetic ketoacidosis, incidence, children, Vojvodina

## Abstract

*Background and Objectives:* The COVID-19 pandemic has led to significant changes globally, which has also affected patients with type 1 diabetes mellitus (T1DM). This study aimed to determine the incidence of T1DM and the characteristics of diabetic ketoacidosis (DKA) during the pandemic comparing it to pre-pandemic period. *Materials and Methods:* Data from patients <19 years with newly diagnosed T1DM between 1st January 2017 and 31st December 2021 from four regional centers in Vojvodina were retrospectively collected and analyzed. *Results:* In 2021, the highest incidence of T1DM in the last five years was recorded, 17.3/100,000. During the pandemic period (2020–2021), there were 99 new-onset T1DM, of which 42.4% presented in DKA, which is significantly higher than in the pre-pandemic period (34.1%). During the pandemic, symptom duration of T1DM lasted significantly longer than before the COVID-19 period. At the age of 10–14 years, the highest incidence of T1DM and COVID-19, the highest frequency rate of DKA, and severe DKA were observed. *Conclusions:* The pandemic is associated with a high incidence rate of T1DM, longer duration of symptoms of T1DM, a high frequency of DKA, and a severe DKA at diagnosis. Patients aged 10–14 years are a risk group for the occurrence of T1DM with severe clinical presentation. Additional studies are needed with a longer study period and in a wider geographical area, with data on exposure to COVID-19 infection, the permanence of new-onset T1DM, and the psychosocial impact of the pandemic.

## 1. Introduction

Since the onset of the COVID-19 pandemic healthcare, institutions have faced challenges adapting their practices to the new circumstances. The preventive epidemiological measures and the fear of infection with severe acute respiratory syndrome coronavirus 2 (SARS-CoV-2) have led to the dominance of a sedentary lifestyle, higher levels of stress, and to avoidance of visits to health facilities. These factors have increased the risk of untimely diagnosis and inadequate treatment of children with T1DM [1,2].

Diabetes mellitus type 1 is an autoimmune disease, where in addition to genetic predisposition, environmental factors also play a role in the etiology, such as exposure to viruses, diet, vitamin D deficiency, reduced diversity of the intestinal microbiome, and increased stress. These trigger factors activate the autoimmune process leading to the destruction of pancreatic β-cells and consequent insulin deficiency by the molecular mimicry mechanism [3,4].

The results of studies on the frequency of T1DM during the COVID-19 pandemic have been controversial. A study in Romania showed an increase in the incidence of T1DM by 16%, from 11.4/100,000 in 2019 to 13.3/100,000 in 2020. It also proved that this increase was significantly higher than the average annual increase in incidence compared to the previous two and a half decades in that country [5]. A study by Unsworth et al. presented data from five regional hospital units in the United Kingdom, collected from late March to early June 2020. In one of the five units, the number of new T1DM cases increased from 2 to 10 and in another from 4 to 10. There were no differences in the other three units compared to the previous five years. However, it showed that many newly diagnosed children presented with severe DKA [6]. Tittel et al. collected data from the Diabetes Prospective Monitoring Registry, from 216 centers in Germany, from mid-March to mid-May 2020. This study showed no significant change in T1DM incidence compared to the predicted incidence based on data from the last decade [7]. Several studies have noted an increased incidence of DKA [8,9] and severe DKA in newly diagnosed T1DM [9,10,11,12,13,14]. It raises the question of whether this frequency and presentation of DKA is a consequence of the high potential of SARS-CoV-2 to initiate the autoimmune process, an inadequate organization of the health care system, or some other reason.

Unsworth et al. state that SARS-CoV-2 shows pathogenicity after binding to the angiotensin-converting enzyme 2 receptor (ACE2) [6]. They referred to the study by Young et al. from 2010, which states that the coronavirus binds to the ACE2 receptor. As a result of the destruction of β-cells, acute diabetes mellitus occurs [15]. However, the exact pathophysiological mechanism of T1DM development after COVID-19 infection is still unknown.

Previous research has included a shorter follow-up period during the COVID pandemic [5,6,7,8,9,10,11,12,13,14], and as far as we know, there are no data on this topic in Serbia. Therefore, we presented the incidence of T1DM and characteristics of DKA in children during and before the pandemic to gain a better insight into the epidemiological data and their association with SARS-CoV-2 infection. A study conducted in Serbia during the pre-pandemic period found that the incidence of T1DM in children and adolescents between 2007 (8.09/100,000) and 2016 (16.31/100,000) doubled, and follow-up and health care planning for these children is necessary [16].

## 2. Materials and Methods

We collected data from four regional centers in Vojvodina: The Institute for Child and Youth Health Care of Vojvodina and the General Hospitals in Subotica, Zrenjanin, and Sombor. The data collection method is explained in detail in a previously published study [16]. The retrospective technique was used to register all patients <19 years of age with newly diagnosed T1DM between 1st January 2017 and 31st December 2021. Data were obtained from patients’ records and the admission and discharge protocol records. Diagnosis of T1DM was made according to the International Society for Pediatric and Adolescent Diabetes (ISPAD) guidelines [17]. Data from medical records were collected on subjects’ gender, date of birth, date of diagnosis, venous pH (bicarbonates when pH was not available), glycemia, HgbA1c, and duration of symptoms. Database forms were distributed to regional centers, where they were completed and returned to the Institute. Patients ≥ 19 years, as well as those with other types of diabetes, were excluded, and duplicates and errors were checked. All participants’ parents or guardians provided informed consent for admission to hospitals and procedures performed during hospitalization. The study protocol was approved by the Institute’s Ethics Committee and was in line with the Declaration of Helsinki.

The study included 231 patients with newly diagnosed T1DM during the period 2017–2021. Patients were divided into the pre-COVID-19 group (2017–2019) and the COVID-19 group (2020–2021) and four age categories: 0–4.9, 5–9.9, 10–14.9, and 15–19 years. The diagnosis and categorization of DKA into mild, moderate, or severe was made according to ISPAD guidelines. Mild DKA was categorized by a pH less than 7.3 or a serum bicarbonate level less than 15 mEq/L; moderate DKA was categorized by a pH less than 7.2 or a serum bicarbonate level less than 10 mEq/L; severe DKA has a pH less than 7.1 and bicarbonate less than 5 mEq/L [18].

The data on new COVID-19 cases in 2020 and 2021 were collected from the Institute of Public Health of Vojvodina Register.

According to the data from the last national census in 2011, there were 387,302 children in the population aged 0–19. These data were used to calculate the incidence rates of T1DM in 2017–2021 and the incidence rates of COVID-19 in 2020–2021 in Vojvodina. The Chi-square and Fisher’s exact test were used to compare the presence of DKA according to factors such as gender, age categories, and the severity of the clinical presentation. Phi correlation coefficient was used to test the relationship between DKA and all mentioned factors. Differences between pre-COVID-19 and COVID-19 DKA frequencies were tested using the One sample Z test. A *t*-test was used to show the differences between the duration of symptoms, glycemia, pH, and HbA1c during the pre-COVID-19 period and COVID-19 period. Values *p* < 0.05 were considered statistically significant. All analyzes were performed using SPSS Version 23.3 (IBM Corp. Released 2015. IBM SPSS Statistics for Windows, Version 23.0. Armonk, NY, USA).

## 3. Results

### 3.1. The Incidence Rate of T1DM during 2017–2021 in Vojvodina

During 2017–2021, there were 231 newly diagnosed cases of T1DM in the pediatric population of Vojvodina, of which 53.2% were male. The highest incidence rate of T1DM was in 2021 in both genders (Table 1 and Figure 1). In the period between 1 January 2020 and 31 December 2021, there were a total of 99 patients with newly diagnosed T1DM.

In 2017–2018, the average increase in the incidence rate of T1DM was 24.36%. In 2018–2019, the average decrease in the incidence rate of T1DM was 21.56%, and in 2019–2020 by 20.04%. In 2020–2021, the average increase in the incidence rate of T1DM was 109.44%.

### 3.2. Frequency of New Cases of T1DM during 2020–2021 per Month

The highest frequency of newly diagnosed T1DM in 2020 was during January and September (15.6%), while the lowest was in June (3.1%) (Figure 2). The highest frequency of new T1DM cases in 2021 was during September (11.9%) and October (13.4%), while the lowest was in June (3.0%). There is seasonality in the appearance of T1DM, with the lowest frequency during the summer months.

### 3.3. The Incidence Rate of T1DM and COVID-19 during 2020–2021 by Age

Figure 3 shows that the highest incidence of T1DM in 2020–2021 was at the age of 10–14 years, while the lowest was in the oldest age group. Furthermore, the highest incidence of COVID-19 in children during 2020–2021 was among 10–14 years of age, while the lowest was in the oldest age group.

### 3.4. Frequency of Newly Discovered T1DM Cases and Characteristics of DKA before Pandemic (2017–2019) and during Pandemic Period (2020–2021)

The highest frequency of new cases of T1DM during 2020–2021 was at the age of 10–14 years, while the lowest frequency was in the youngest age group. During pre-pandemic years the highest frequency of new cases of T1DM was in the 5–14 age group, and the lowest frequency was in the oldest age group.

At the time of diagnosis, 42.4% of patients were in a state of DKA during pandemic years, most frequently in severe DKA, while 57.6% were in a compensated state without DKA. In pre-pandemic years 34.1% of patients presented in a state of DKA. One sample Z test showed that DKA frequency during the pre-COVID-19 period 2017–2019 (45/132; 34.10%) is statistically different from DKA frequency during the period 2020–2021 (42/99; 42.42%) (z = −6.28; *p* < 0.001).

In pandemic years the highest frequency of DKA was at the age of 10–14 years, then at the age of 5–9 years, with the youngest age group having the lowest frequency. The Chi-square test showed no statistically significant difference in the presence of DKA between different age groups (χ2 = 1.65; df = 3; *p* = 0.65). The highest incidence of severe DKA was in the age group of 10-14 years, and the lowest was at 15–19 years. Fisher’s exact test showed that the severity of the clinical presentation is not related to age (Fisher’s exact test = 6.216; df = 9; *p* = 0.734).

In the pre-pandemic period, the Chi-square test showed no statistically significant difference in the presence of DKA between different age groups (χ2 = 5.424; df = 3; *p* = 0.144).

New cases of T1DM during the pandemic were in a higher percentage of male than female patients. A high percentage of patients of both genders were in the DKA state. A higher percentage of male patients presented in DKA compared to females. Fisher’s exact test did not show any significant association between DKA and gender (Fisher’s exact test = 0.53; df = 1; *p* = 0.54). A higher percentage of male patients presented with moderate and severe DKA than female patients, while a higher percentage of female patients presented with mild DKA. Chi-square test showed no difference in the severity of the clinical presentation by gender (χ2 = 3.650; df = 3; *p* = 0.315).

In the pre-pandemic period, Fisher’s exact test did not show any significant association between the overall number of DKA and gender (Fisher’s exact test = 0.169; df = 1; *p* = 0.715). As it is for the severity of the clinical presentation, Fisher’s exact test showed a statistically significant difference in males and females (χ2 = 7.883; df = 3; *p* = 0.045). Data are shown in Table 2. Phi correlation coefficient showed a small positive association between severity of DKA and gender. Males had a higher number of moderate and severe DKA than females in the pre-COVID-19 period (Phi = 0.244; *p* = 0.043).

A comparison of mean values of factors that are correlated with DKA in pre-pandemic and pandemic years showed that the duration of symptoms of T1DM during the COVID-19 period (24.19 ± 16.25 days) lasted statistically significantly longer than symptoms of T1DM before COVID-19 period (16.16 ± 15.81 days) (*t*-test = −2.36; *p* = 0.019). Regarding the average values of the remaining factors, glycemia, pH, and HbA1c during the pre-COVID-19 period and COVID-19 period, no statistically significant difference was found between the COVID-19 and the pre-COVID-19 period.

## 4. Discussion

This study is the first report on the incidence rate of T1DM and the characteristics of DKA during the pandemic in Vojvodina. The incidence of T1DM in 2021 was the highest in the last five years. When compared with the results in Serbia in the pre-pandemic period [16], this is the highest incidence of T1DM in 15 years. In neighboring Romania in 2019–2020, there was an increase in incidence [5], while we observed a decrease in incidence in the same period. In a study in Canada, there was no change in the incidence of T1DM before and during the pandemic [12]. The high incidence in Vojvodina can be hypothetically related to changes in environmental factors and people’s lifestyles. The first year of the pandemic was characterized by strict epidemiological measures, with the closure of all educational and most public institutions over a long period. In the following year, these institutions worked for most of the year. As a result, there was more direct contact and a higher chance of infection with SARS-CoV-2 and other viruses. Reduced physical activity due to isolation and increased stress were significant contributing factors to the development of T1DM [1].

During the pandemic, a statistically significantly higher frequency of patients with DKA (42.42%) in Vojvodina was observed compared to the period before. In Serbia in the pre-pandemic period, there was a high frequency of DKA (35.1%) at the time of T1DM diagnosis [16,19,20]. DKA frequency during the period 2007–2017 in Vojvodina (147/407; 36.12%) is noticeably different from DKA frequency during the COVID-19 period 2020-2021 (42/99; 42.42%) [16]. A high frequency of severe DKA (15.2%) was also observed during the pandemic in our study. According to research in the United Kingdom, a higher frequency of DKA was registered during the pandemic than before [13]. A more severe clinical presentation of new cases of T1DM, with the high incidence of DKA during the pandemic, was also reported in the United States compared to previous years [8]. In a study in Canada, in addition to a higher incidence of DKA and severe DKA, there was a significant increase in admission to pediatric intensive care for newly diagnosed T1DM patients in 2020 compared to the pre-pandemic period [12]. In a large hospital in Southeastern Brazil, research was conducted between April and August 2017, 2018, and 2019 (the pre-pandemic period) and between April and August 2020 (the pandemic period). The higher frequency and severity of DKA and a higher number of new cases of T1DM were observed in children and adolescents during the pandemic [9]. A study in Finland compared T1DM and DKA characteristics before and during the pandemic. No antibodies to SARS-CoV-2 were present in patients with newly diagnosed T1DM during the pandemic. This study was conducted during a period of low incidence of COVID-19, with an insufficient increase in the incidence of T1DM, which could explain the incidence of DKA. Researchers stated that a higher incidence of severe DKA is not a direct consequence of COVID-19 but a delay in diagnosis [14]. We also agree that the high percentage of patients with severe DKA indicates a delay in the diagnosis. In addition, T1DM symptoms lasted longer in the pandemic than in the pre-pandemic period in our research. The potential causes may be avoiding visiting health care institutions due to fear of viral transmission or insufficient parent’s care for the children’s health as a consequence of preoccupation with the pandemic. On the other hand, other causes could be increased use of telemedicine, inadequate organization or reduced availability of health services, or overlap of T1DM symptoms with a viral infection. DKA, especially severe DKA, is the leading cause of acute mortality and brain edema in children with T1DM [18]. Additional preventive and educational measures for patients and parents, re-education of health workers, and reorganization of health services are needed. Health education programs should be implemented in schools and preschool institutions, live or via a virtual platform on the subjects of T1DM, DKA, and COVID-19. Furthermore, encourage the involvement of the volunteer community in implementing the health education program. In order to increase the availability of health care, it is necessary to ensure the optimal number of workers in health institutions, better organize staff and optimize work performance. In the case of socioeconomic or spatial obstacles, it is necessary to enable better accessibility of telemedicine to achieve faster contact with medical staff. Educate healthcare workers on how to communicate more effectively with patients and with each other. Provide additional funds for the implementation of telemedicine, equipment, and accompanying costs. It is necessary to educate health workers on the subject of T1DM and DKA at all levels of health care to recognize the symptoms of T1DM earlier and avoid unnecessary laboratory analyses. Future studies should determine parents’ attitudes and access to health services during the pandemic.

At the age of 10–14 years, the highest incidence rate of T1DM was followed by the highest incidence rate of COVID-19. The same association was in the oldest age group, with the lowest incidence rate. This correlation suggests that SARS-CoV-2 may have a potentially diabetogenic effect. It is known that viruses can activate the autoimmune process in T1DM [3]. We hypothesize that SARS-CoV-2 could be a trigger and lead to pancreatic β-cells destruction, insulin deficiency, and T1DM. It is currently unknown when precisely T1DM develops following COVID-19 infection, what the exact pathophysiological mechanism is, and whether newly diagnosed T1DM is transient or permanent. These issues should be addressed in future research.

The highest incidence rate of newly diagnosed T1DM during the pandemic was in the age group of 10–14 years. The lowest incidence rate was in the oldest group, and their presentation of DKA was usually mild. These results correlate with the results of research by Vukovic et al. during the pre-pandemic period [16]. In Canada, the highest incidence of T1DM in 2020 was in the age group 6–11 years [12]. In our study, during the pandemic, the highest frequency of DKA and severe DKA was in patients aged 10–14 years, while the lowest frequency of DKA was in the youngest age group. It differs from previous studies, where the highest incidence of DKA and significantly more severe DKA were in children younger than five years [16,19]. According to ISPAD, children younger than two years are at the highest risk for developing DKA in newly diagnosed T1DM [18]. Vaccination against COVID-19 began in the second year of the COVID pandemic, first at 16-year-olds. The youngest and oldest children attended classes through online platforms during the pandemic, while children aged 10–14 mostly went to school every day and had multiple contacts. Therefore, it is necessary to prevent interpersonal contact and reduce viral transmission. Contrary to previous understandings [16,18,19], our research indicates a need to form a strategy and implement preventive measures, focusing on the age of 10–14 years.

We showed the incidence of T1DM in the last five years and the characteristics of DKA during two years. That is a relatively short time to make detailed and exact conclusions about the impact of the COVID-19 pandemic. There are no data on the presence of antibodies to SARS-CoV-2 and previous COVID-19 infection in patients with newly diagnosed T1DM to more accurately determine the association between T1DM and SARS-CoV-2. There is also a lack of information on the impact of the COVID-19 pandemic on children’s lifestyles. Future studies should include this data, the permanence of new-onset T1DM, and a longer research period.

Since this is a multicenter study, the data refer to the entire territory of Vojvodina, which is an advantage of this research. Furthermore, a centralized type of health system enables better supervision and direct communication of pediatric endocrinologists. Moreover, it contributes to the accuracy of the data. Previously studies in Serbia had the same methodological principles [16,19,20], which allowed us better insight into the current epidemiological characteristics. No studies have been published on this topic in Serbia, which cover a period of two years during the COVID-19 pandemic, which is a significant addition to the literature so far.

## 5. Conclusions

Our study indicates that the COVID-19 pandemic is associated with a longer duration of symptoms than in the previous period, high incidence of T1DM, a high frequency of DKA, and severe DKA at diagnosis. Patients aged 10–14 years are a risk group for the occurrence of T1DM with severe clinical presentation. Additional preventive and educational measures for patients, parents, and health workers and the reorganization of health services are needed. Additional research is required, covering a longer study period, a wider geographical area, data on the presence of COVID-19 infection, the permanence of new-onset T1DM, and the psychosocial impact of the pandemic.

## Figures and Tables

**Figure 1 medicina-58-01013-f001:**
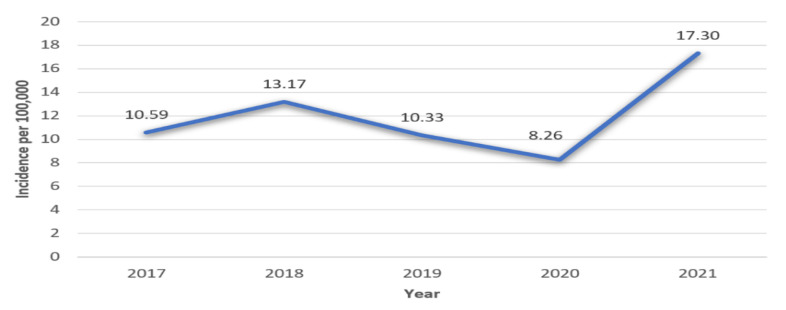
Incidence rates (per 100,000) of T1DM during 2017–2021 in Vojvodina.

**Figure 2 medicina-58-01013-f002:**
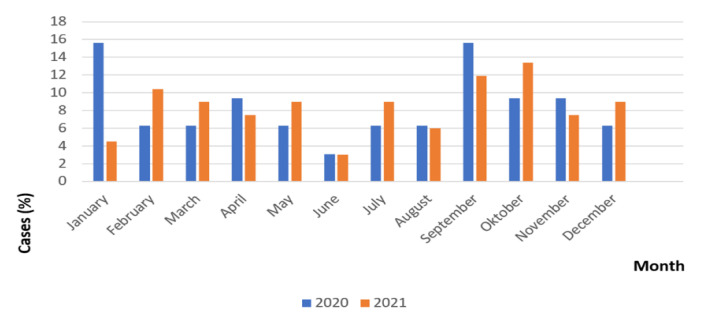
Frequency of new cases of T1DM during 2020–2021 by months.

**Figure 3 medicina-58-01013-f003:**
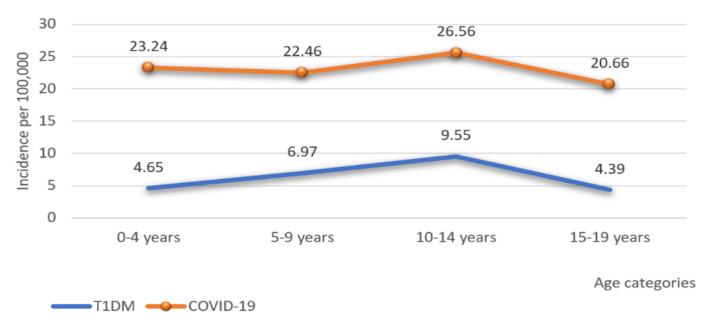
The incidence rate of T1DM and COVID-19 during 2020-2021 by age.

**Table 1 medicina-58-01013-t001:** Incidence rates (per 100,000) of T1DM by gender during 2017–2021 in Vojvodina.

Year	Females Incidence	Males Incidence
2017	4.39	6.20
2018	6.45	6.71
2019	4.39	5.94
2020	3.61	4.65
2021	9.03	8.26

**Table 2 medicina-58-01013-t002:** Presenting state at diagnosis by age categories and gender during 2017–2021.

	Compensated (*n*,%)		DKA								Total (*n*,%)	
			Mild (*n*,%)		Moderate (*n*,%)		Severe (*n*,%)		Total DKA (*n*,%)			
Age (Years)	Pre-COV	COV	Pre-COV	COV	Pre-COV	COV	Pre-COV	COV	Pre-COV	COV	Pre-COV	COV
0–4	9 (6.8%)	10 (10.1%)	2 (1.5%)	2 (3.0%)	2 (1.5%)	3 (3.0%)	5 (3.8)	3 (3.0%)	9 (6.8%)	8 (8.2%)	18 (13.6%)	18 (18.2%)
5–9	34 (25.8%)	15 (15.2%)	9 (6.8%)	4 (4.0%)	3 (2.3%)	5 (5.0%)	4 (3.0%)	3 (3.0%)	16 (12.1%)	12 (12.1%)	50 (37.9%)	27 (27.2%)
10–14	25 (18.9%)	24 (24.2%)	7 (5.3%)	4 (4.0%)	3 (2.3%)	2 (2.1%)	6 (4.5%)	7 (7.1%)	16 (12.1%)	13 (13.1%)	41 (31.1%)	37 (37.4%)
15–19	19 (14.4%)	8 (8.1%)	1 (0.8%)	4 (4.0%)	2 (1.5%)	3 (3.0%)	1 (0.8%)	2 (2.1%)	4 (3.0%)	9 (9.1%)	23 (17.4%)	17 (17.2%)
**Total**	87 (65.9%)	57 (57.6%)	19 (14.4%)	14 (14.1%)	10 (7.6%)	13 (13.1%)	16 (12.1%)	15 (15.2%)	45 (34.1%)	42 (42.4%)	132 (100%)	99 (100%)
Gender												
Females	40 (30.3%)	30 (30.3%)	12 (19.1%)	9 (9.%)	1 (0.8%)	5 (5.1%)	6 (4.5%)	5 (5.1%)	19 (14.4%)	19 (19.2%)	59 (44.7%)	49 (49.5%)
Males	47 (35.6%)	27 (27.3%)	7 (5.3%)	5 (5.1%)	9 (6.8%)	10 (10.1%)	10 (7.6%)	10 (10.1%)	26 (19.7%)	23 (23.2%)	73 (55.3%)	50 (50.5%)

## Data Availability

The data presented can be provided on request.

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
