# Peer review of "Incidence of Type 1 Diabetes Mellitus and Characteristics of Diabetic Ketoacidosis in Children and Adolescents during the First Two Years of the COVID-19 Pandemic in Vojvodina"

_medicina, 2022, doi:10.3390/medicina58081013_

Round 1
Reviewer 1 Report
I read with great interest the manuscript entitled “Incidence of type 1 diabetes mellitus and characteristics of 1 diabetic ketoacidosis in children and adolescents during the 2
first two years of the COVID-19 pandemic in Vojvodina”. The study gives insight about the global incidence of T1DM and DKA in children during the COVID 19 pandemic.
Comments:
Introduction:
· Line 20: outbreak of the COVID-19 pandemic. Please remove the outbreak as COVID 19 is now a pandemic not an outbreak.
Materials and Methods:
· It would be nice to compare the incidence of DKA before and during the COVID 19 pandemic and to compare the factors associated with it like age, gender, BMI, HbA1C, socioeconomic standard, health care accessibility, stress and depression.
Results:
· I would suggest to add a correlation for factors correlated with DKA during the covi19 pandemic.
· It would be nice to add multivariate regression analysis for the factors associated with DKA during the covid19 pandemic.
Discussion:
· The discussion is rather narrative. It is better to discuss the study findings with the relevant literature.
Author Response
"Please see the attachment."

Reviewer 2 Report
For the article “Incidence of type 1 diabetes mellitus and characteristics of diabetic ketoacidosis in children and adolescents during the first two years of the COVID-19 pandemic in Vojvodina” Authors Ivana Vorgučin * , Marijana Savin , Đurđina Stanković * , Dejan Miljković, et al
(Manuscript ID medicina-1810907)
Article is devoted to important and interesting topic – incidence of Type 1 Diabetes Mellitus in children population (up to age 19). Results have descriptive epidemiologic characteristics. Main findings are:1) notable increase of T1DM incidence rate during the Covid-19 pandemic (year 2020) in comparison with pre-pandemic period (2017-2019), what is leading to conclusion about necessity of more profound pathogenetic explanatory studies on individual level; 2) increase of frequency and severity of diabetic ketoacidosis, what leads to conclusion about improvements in the problem management.
There is no need for major revisions. Also, technically there are no remarks. One may wander about expression of incidence rate per 100 000 not mentioning is this population or person-years. As academically correct version is “per person-years”, it is common practice and acceptable not to mention “person-years” to ease readability for clinicians.
Only remark for minor revision is related just to one part of sentence in discussion part (line 200) and conclusions: "... reorganization of health services are needed". Readers may wish to understand what kind of reorganizations authors mean as it is not obvious how this can change outcome. Explanation could be added to the discussion.
Author Response
Dear Reviewer,
Thank you very much for your favorable review and constructive suggestions. We have read the comments and suggestions very carefully, and revised our manuscript accordingly. Corrections can be seen in the revised manuscript Word document.
The followings are our point-by-point responses to the reviewers’ comments and suggestions; we hope that the revised manuscript now meets the requirements for the caliber of the journal Medicina.
We are looking forward to hearing from you soon. Thank you very much for your attention.
Very Sincerely,
Authors
A point-by-point response to Reviewer #2
Article is devoted to important and interesting topic – incidence of Type 1 Diabetes Mellitus in children population (up to age 19). Results have descriptive epidemiologic characteristics. Main findings are:1) notable increase of T1DM incidence rate during the Covid-19 pandemic (year 2020) in comparison with pre-pandemic period (2017-2019), what is leading to conclusion about necessity of more profound pathogenetic explanatory studies on individual level; 2) increase of frequency and severity of diabetic ketoacidosis, what leads to conclusion about improvements in the problem management.
There is no need for major revisions. Also, technically there are no remarks.
- One may wander about expression of incidence rate per 100 000 not mentioning is this population or person-years. As academically correct version is “per person-years”, it is common practice and acceptable not to mention “person-years” to ease readability for clinicians.
The denominator used in the formula for incidence represents the number of children in the population aged 0-19 years. The data was taken from the last national census in 2011 and was used as a calculated number of this population during the whole time interval of our study (2017-2021).
- Only remark for minor revision is related just to one part of sentence in discussion part (line 200) and conclusions: "... reorganization of health services are needed". Readers may wish to understand what kind of reorganizations authors mean as it is not obvious how this can change outcome. Explanation could be added to the discussion.
We have added an explanation to the discussion: Additional preventive and educational measures for patients and parents, re-education of health workers, and reorganization of health services are needed. Health education programs should be implemented in schools and preschool institutions, live or via a virtual platform on the subjects of T1DM, DKA, and COVID-19. Also, encourage the involvement of the volunteer community to implement the health education program. To increase the availability of health care, it is necessary to ensure the optimal number of workers in health institutions, better organize staff and optimize work performance. In the case of socioeconomic or spatial obstacles, it is necessary to enable better accessibility of telemedicine to achieve faster contact with healthcare workers. Educate healthcare workers on how to communicate more effectively with patients and with each other. Provide additional funds for the implementation of telemedicine, equipment, and accompanying costs. For healthcare professionals to recognize the symptoms of T1DM earlier and to avoid unnecessary laboratory analyses, it is necessary to educate on the subject of T1DM and DKA at all levels of healthcare.
This manuscript is a resubmission of an earlier submission. The following is a list of the peer review reports and author responses from that submission.